🔓 | **Open Peer Review** | Clinical Microbiology | Research Article

# Genetic characteristics and risk factors of extended-spectrum β-lactamase-producing *Escherichia coli* in Chinese intensive care unit: a prospective molecular epidemiology study

Ying Ding,[1,2,3] Meijun Song,[2,3,4] Yi Yang,[5] Hemu Zhuang,[2,3,4] Qiuxiang Pan,[6] Lijie Xu,[1,2,3] Jintao He,[2,3,4] Junxin Zhou,[2,3,4] Haiyang Liu,[2,3,4] Yongjun Lin,[1] Min Liang,[1,2,3] Xiuliu Guo,[1,2,3] Yunsong Yu,[2,3,4] Yan Jiang[2,3,4]

**ABSTRACT**   In this 4-month-long prospective observational study, we explored the colonization rate of extended-spectrum β-lactamase-producing *Escherichia coli* (ESBL-EC) in the patient and ward environment of an intensive care unit (ICU). Additionally, we evaluated the risk factors for colonization and analyzed genomic characteristics and modes of transmission of isolates. Clinical samples were collected from patients and the environment to isolate and screen *E. coli* strains. ESBL-EC from the *E. coli* strains was identified using ESBL confirmation and antibiotic susceptibility tests and subsequently characterized using whole-genome sequencing. Clinical data were collected and further analyzed. Among the 214 *E. coli* isolates, 82 were ESBL-EC, with CTX-M-14 being the dominant enzyme, followed by CTX-M-55 and CTX-M-15. The predominant sequence types (STs) among the 82 ESBL-EC strains were ST10, followed by ST131 and ST1193. Using multiple logistic regression, exposure to third-generation cephalosporins and a special class of anti-positive-bacterial drugs, as well as albumin and enteral nutrition, were high-risk factors for ESBL-EC colonization. The clonal transmissions of ESBL-EC in the ICU were predominantly attributed to the movement of healthcare workers. More effective interventions and active screening are needed to prevent and control ESBL-EC colonization.

**IMPORTANCE**   The increasing prevalence of extended-spectrum β-lactamase-producing *Escherichia coli* (ESBL-EC) has made drug-resistant bacterial infections rise, endangering people's health and causing socioeconomic burdens. We conducted an ESBL-EC screening program for patients and ward environments in an intensive care unit (ICU). The aim was to describe the molecular characteristics of ESBL-EC and the risk factors for ESBL-EC colonization. In our hospital, the colonization rate of ESBL-EC remained high. The dominant sequence type was ST10, which might be considered a strain of notable concern, possibly causing future outbreaks. However, ST131 and ST1193 should also be considered because they were associated with the majority of the ESBL-EC isolates found. Notably, CTX-M-14 gene screening should be considered in medication guidance because it is the main ESBL enzyme. Owing to the high transmission rate of ESBL-EC, effective interventions and active screening are critical for preventing and controlling its spread, guiding clinicians in rational antibiotic use.

**KEYWORDS**   ICU, ESBL-EC, clonal transmission, antimicrobial resistance

Extended-spectrum β-lactamase-producing *Escherichia coli* (ESBL-EC) has currently become an emerging public health concern. ESBLs were functionally defined as a series of β-lactamases that are able to hydrolyze penicillins, first-, second-, and third-generation cephalosporins, and monobactams (aztreonam) but are unable to hydrolyze

**Peer Reviewer** Kai Zhou, Shenzhen University, Shenzhen, Guangdong, China

Address correspondence to Yan Jiang, jiangy@zju.edu.cn, or Yunsong Yu, yvys119@zju.edu.cn.

Ying Ding and Meijun Song contributed equally to this article. Author order was determined in order of increasing seniority.

The authors declare no conflict of interest.

See the funding table on p. 13.

cephamycins or carbapenems. ESBL-EC can be inhibited by β-lactamase inhibitors such as clavulanic acid (1). However, the definition of this enzyme was still expanding, owing to the discoveries of novel ESBLs (2).

Over the past decade, the carriage rate of ESBL-producing bacteria has been steadily increasing both in the community and in hospitals due to the pandemic of CTX-M enzyme (one of the major types of ESBLs) (3–5). International travel was a potential risk factor for the inter-regional spread and colonization of ESBL-producing bacteria (6). Furthermore, reports from epidemiological studies showed that poverty, high population density, and water pollution were the main factors promoting ESBL-producing *Enterobacteriaceae* spread (3), suggesting that this ESBL-producing *Enterobacteriaceae* infection may be a more serious threat to the healthcare system of developing than developed countries. Recently, studies reported that the ESBL-producing strain was mainly derived from *Klebsiella pneumoniae* or *E. coli* (7). Notably, an increased ESBL-producing bacterial infection is associated with a serious health and economic burden. Several retrospective cohort studies showed that the bloodstream infection caused by ESBL-producing or third-generation cephalosporin-resistant *Enterobacteriaceae* increased mortality, improved healthcare costs, and prolonged the length of hospital stay of patients (8). It is estimated that infection caused by ESBL-EC increase the risk of clinical failure and mortality, and also with an additional cost in Canada (9).

There was rising concern regarding the hospital-acquired infection of ESBL-producing organisms, especially in long-term care facilities (10). Long hospital stays, invasive medical device use, recent antibiotic use, history of surgery or invasive procedures, and hemodialysis have all been reported as risk factors that contribute to the spread of this infection in healthcare settings (1, 10, 11). Studies reported that third-generation cephalosporin, aztreonam, trimethoprim-sulfamethoxazole, aminoglycosides, and metronidazole use were associated with this infection (12–15). However, whether critically ill patients benefit from screening for intestinal carriage of ESBL-producing *Enterobacteriaceae* remains unclear (10, 16). Furthermore, there is a relative lack of large-scale screening studies on ESBL-EC.

Considering its high prevalence among patients in intensive care units (ICUs) in China, we conducted a prospective observational study to explore the colonization rates, genomic characteristics, and risk factors of ESBL-EC in patients and the environment in the ICU. These findings might be beneficial for preventing the spread of ESBL-EC in the ICU and guiding clinicians in the rational use of antibiotics.

## MATERIALS AND METHODS

### Strain collection

We conducted a prospective study to explore the colonization and dissemination of ESBL-EC in the ICU environment. The study was conducted between 1 April 2021 and 31 July 2021, with screenings performed every Tuesday morning. This study was conducted in the 28-bed ICU of Sir Run Run Shaw Hospital, Zhejiang University School of Medicine. We obtained clinical samples from all patients in the ICU weekly. The collected samples included throat and anal swabs and samples from their endotracheal, gastric, tracheostomy, and nasointestinal tubes. Patients' beds and their surrounding equipment, including bed rails, ventilator button panels, electrocardiography monitors, micropumps, on/off buttons, nebulizers, stethoscopes, medical pendants, bed regulators, bedside tables, computer mouses and keyboards, loop hooks for intravenous infusion poles, treatment trolleys, and room lockers, were all sampled. Furthermore, we collected samples weekly from all sinks in the wards as well as from the interior surfaces of drains, faucet surfaces, sink countertops, interior walls of overflows, and water in pipes.

### Sampling and strain identification

The sampling personnel used Copan swabs to obtain samples from patients, bed units, and sinks. The swabs were subsequently placed in tryptic soy broth and cultured

overnight. Approximately 20 µL solution of the culture was subjected to CHROMagar *E. coli* screening plates (CHROMagar, Paris, France) and cultured overnight at 37°C. A single colony was chosen from each plate and inoculated onto a Mueller-Hinton agar plate. Subsequently, the bacterial division and ring were marked, and the plates were cultured overnight at 37°C. On the third day, a single colony from each plate was picked for matrix-assisted laser desorption ionization-time of flight mass spectrometry (bioMérieux, Marcy-l'Etoile, France) to determine colonies that were to *E. coli* strains.

## Antimicrobial susceptibility testing

The minimum inhibitory concentrations (MICs) of amikacin, meropenem, imipenem, ertapenem, piperacillin/tazobactam, fosfomycin, cefepime, ciprofloxacin, aztreonam, ceftazidime, and ceftriaxone were measured using the agar dilution method. MIC breakpoints were interpreted using the Clinical and Laboratory Standards Institute (CLSI) guidelines (2020 edition) (17). Standard strains of *E. coli* ATCC 25922 were used as the positive control.

## ESBL phenotypic confirmation

For confirmation of ESBL, a combined disc test was performed using cefotaxime, cefotaxime/clavulanic acid, ceftazidime, and ceftazidime/clavulanic acid following the CLSI guidelines (2020 edition) (17). An increase in the zone of inhibition by ≥5 mm for either cefotaxime/clavulanic acid compared with cefotaxime alone and ceftazidime/clavulanic acid with ceftazidime alone was interpreted as confirmed ESBL (18). Standard strains of *E. coli* ATCC 25922 were used as the positive control.

## Clinical data collection

The clinical data of patients were collected between 1 April 2021 and 31 July 2021. The following were the inclusion criteria: (i) ESBL-EC colonization can be detected in any of six parts: anal swab, tracheal intubation, oral cavity, gastric tube, nasointestinal tube, and tracheotomy. (ii) Patients hospitalized in ICU. The exclusion criteria included cases with serious missing clinical data. Finally, 159 eligible cases were included. The clinical data of patients who met the inclusion criteria were collected. The data included age, gender, body mass index (BMI), length of hospitalization before admission to ICU, chronic kidney disease, hypertension, diabetes, antibiotic use of 3 months before sampling (including use of enzyme inhibitors, carbapenems, quinolones, and special-positive bacteria drugs), use of immunosuppressants, malignant tumor, history of operation within 1 year, indwelling catheter, a central venous catheter, enteral nutrition. This study was reviewed and approved by the Ethics Committee of Sir Run Run Shaw Hospital, Zhejiang University School of Medicine (No. 20201217-33).

## Whole-genome sequencing and bioinformatic analysis

The whole-genome sequencing and bioinformatic analysis followed the same procedure as in our previous study (19). Briefly, the DNA was sequenced using the Illumina HiSeq (Illumina, San Diego, CA, USA). The raw data of sequencing were assembled by the Shovill pipeline (version 4.4.5, https://github.com/tseemann/shovill). The multilocus sequence typing (MLST) used in the survey was the Achtman scheme, which was performed using mlst (https://github.com/tseemann/mlst) with the PubMLST data set (20). Using ABRicate (v0.8.13, https://github.com/tseemann/abricate), we screened resistance genes referring to the National Center for Biotechnology Information AMRFinderPlus database (21). The single nucleotide polymorphism (SNP) was calculated using the snippy pipeline (version 4.4.5, https://github.com/tseemann/snippy). We used the IQ-TREE pipeline (version 2.1.2) to generate the maximum likelihood tree (22). An SNP distance ≤22 was set as the criterion for indicating that the two strains were homologous (23).

## Statistics

The patients' clinical database was created using Excel 2022, and the data were analyzed using the SPSS Software (V22.0, SPSS, Inc., Chicago, IL, USA). Continuous variables following a normal distribution are expressed as the mean ± standard deviation, and categorical variables are presented as percentages. An unpaired *t*-test was used to compare the means between the two groups. A chi-squared test was used to compare proportions between two groups, whereas the Mann–Whitney *U*-test was used to compare medians. Univariate binary logistic regression was used to determine potential risk factors for the ESBL-EC spread. All potential covariates determined using univariate analysis ($P < 0.05$) were subsequently entered into the binary logistic multiple regression model. A *P* value of <0.05 was considered statistically significant.

## RESULTS

### Molecular of ESBL-EC

The initial sampling of this study was conducted on 1 April 2021. From this week onward, we sampled patients and the environment every Tuesday for 17 weeks until 31 July 2021. Overall, samples were obtained from 159 patients in the ICU at least once. A total of 214 *E. coli* strains were isolated, of which 191 were isolated from 99 different patients and 23 were isolated from the surrounding environment of patients in the ICU ward. We detected *E. coli* in 96 of these patients, whereas ESBL-EC was only detected in 41 of them. Patients with ESBL-EC in their throat and anal swabs, as well as in samples from their endotracheal, gastric, tracheostomy, and nasointestinal tubes, were ESBL-EC carriers. The positive rate of ESBL-EC carriers was 42.71% (41/96). Furthermore, 191 out of 1,140 non-duplicate samples from 159 patients were positive for *E. coli*, and the detection rate was 16.75% (191/1140). The samples were mainly isolated from the anal swabs, which accounted for 90.05% (172/191). Of these 191 non-duplicate samples, 75 had ESBL-EC, with an ESBL-positive rate of 39.27% (75/191). Meanwhile, 23 out of 5,824 environmental samples were positive for *E. coli*, and the detection rate was 0.39% (23/5824). These samples were mainly isolated from the bed railing, which accounted for 21.74% (5/23). Seven ESBL-EC strains of the above 23 environmental samples were detected, and the positive rate was 30.43% (7/23) (Fig. 1).

In this study, patients who were found with ESBL-EC colonization after arriving at the ICU within 48 h were categorized as non-ICU-acquired. However, those in whom ESBL-EC colonization was not detected after the first ICU screening within 48 h of arrival and subsequently were found to be ESBL-EC positive were categorized as ICU-acquired. ESBL-EC that was detected after 48 h for patients entering the ICU who were initially ESBL-EC negative during the first ICU screening was defined as uncertain ICU-obtained

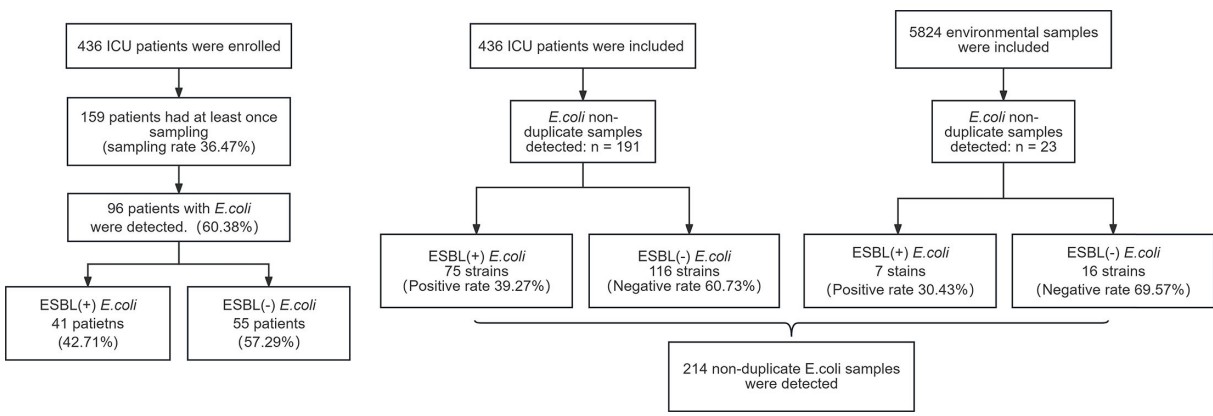

FIG 1  The strain screening method for clinical samples in this study. Strains isolated from oropharyngeal swabs, rectal swabs, gastric tubes, nasointestinal tubes, tracheal intubation tubes, and tracheotomy tube swabs from the same patient were defined as non-duplicate samples, while strains isolated from the same swab type from the same patient in the same week were defined as duplicate samples.

ESBL-EC. Based on the history of patients admitted to ICU, 17 were considered to have ICU-acquired ESBL-EC, and the colonization rate was 41.46% (17/41). Of the 17 patients, four cases were acquired through ICU transmission, accounting for 23.53% (4/17). Eight patients were considered non-ICU-acquired, and the colonization rate was 19.51% (8/41). Meanwhile, for 17 patients (39.02%, 16/41), it was unclear whether they had ICU- or non-ICU-acquired ESBL-EC.

## Risk factors for ESBL-EC colonization

We used univariate analysis to analyze the clinical data of patients admitted to the ICU during the study period to identify risk factors for ESBL-EC colonization. We found that a history of surgery within 1 year was the high-risk factor for ESBL-EC colonization ($P < 0.05$) (Table 1). All risk factors were included in the logistic regression model for multivariate analysis. The results further showed clarifications that enteral nutrition (OR = 4.358, 95% CI = 1.193–18.799, $P = 0.034$), history of third-generation cephalosporin exposure (OR = 25.054, 95% CI = 2.206–698.842, $P = 0.020$), history of special class of anti-positive bacteria drugs exposure (OR = 6.717, 95% CI = 1.309–42.892, $P = 0.030$), and albumin (OR = 0.847, 95% CI = 0.717–0.983, $P = 0.036$) were high-risk factors for ESBL-EC colonization (Table 2).

## Antimicrobial resistance of ESBL-EC

The resistance rates of the 82 ESBL-EC strains to amikacin, meropenem, and imipenem were all lower than 10%. The drug resistance rates of ertapenem, piperacillin/tazobactam, and fosfomycin were 12.2%, 23.17%, and 26.83%, respectively. The ESBL-EC strains had a high resistance rate to cefepime (91.46%), ciprofloxacin (89.02%), aztreonam (76.83%), and ceftazidime (63.41%. In addition, the resistance rates of amikacin, piperacillin/tazobactam, fosfomycin, ceftazidime, aztreonam, ciprofloxacin, cefepime, and ceftriaxone in the ESBL-EC group were higher than in the non-ESBL-EC group (Fig. 2).

## ESBL enzyme distribution of ESBL-EC

The analysis of ESBL type showed that all 82 ESBL-EC strains produced CTX-M enzymes (100%, 82/82), as shown in Fig. 3. Among them, there were 38 strains of CTX-M-1 cluster, including 17 strains producing CTX-M-55 (44.74%, 17/38), 12 strains producing CTX-M-15 (31.58%, 12/38), and 8 strains producing CTX-M-3 (21.05%, 8/38) enzymes, as well as 1 strain producing CTX-M-64 (2.63%, 1/38) enzyme. There were 50 strains producing the CTX-M-9 cluster, including 22 strains producing CTX-M-14 (44%, 22/50), 11 strains producing CTX-M-27 (22%, 11/50), 11 strains producing CTX-M-65 (22%, 11/50), and 6 strains producing CTX-M-9 enzyme (12%, 6/50). In addition, there were nine strains of ESBL-EC producing other clusters, including eight strains producing OXA-1 enzyme (9.76% of ESBL-EC, 8/82) and one strain producing TEM-20 enzyme (1.22%, 1/82). The CTX-M-14 originating from the CTX-M-9 cluster was the main ESBL enzyme in ESBL-EC strains isolated from our ICU.

## MLST distribution of ESBL-EC

We identified 25 different ST types from the 82 ESBL-EC strains using the MLST analysis, among which ST10 (17.07%, 14/82) was the predominant clone. The following STs were prevalent among the ESBL-EC strains: ST131 (10), ST1193 (10), ST38 (6), and ST95 (6). The remaining STs observed were all less than five strains. Phylogenetic analysis was used to divide the 88 ESBL-EC isolates into three clades: clade A, containing 21 strains, mainly ST10; clade B, containing 7 strains, mainly ST38; and clade C, containing 26 strains, mainly ST131 and ST1193 (Fig. 4). Among the strains we isolated, ST10 was the most prevalent, and all of them were clustered in clade A.

**TABLE 1** High-risk factors for bacterial colonization of ESBL-EC in ICU patients

| | | Non-ESBL-EC | ESBL-EC | P |
|---|---|---|---|---|
| Sex | Male | 32 | 30 | 0.192 |
| | Female | 23 | 11 | |
| Age (years) | | 69,57,78, | 71,57,80, | 0.773 |
| BMI (kg/m$^2$) | | 22.49 ± 3.91 | 23.63 ± 4.93 | 0.238 |
| Albumin (g/L) | | 31.03 ± 4.93 | 29.39 ± 3.99 | 0.076 |
| Length of stay before ICU (days) | | 0, 0, 7.5 | 0, 0, 5 | 0.294 |
| Chronic kidney diseases | Yes | 6 | 2 | 0.460 |
| | No | 49 | 39 | |
| Immunosuppressive agents | Yes | 9 | 6 | 1 |
| | No | 46 | 35 | |
| Cancer | Yes | 11 | 9 | 1 |
| | No | 44 | 32 | |
| History of surgery within 1 year | Yes | 13 | 19 | 0.034[a] |
| | No | 42 | 22 | |
| Catheterization | Yes | 44 | 31 | 0.791 |
| | No | 11 | 10 | |
| Antibiotic exposure within 3 months | | | | |
| Third generation of cephalosporin | Yes | 1 | 4 | 0.160 |
| | No | 54 | 37 | |
| Enzyme inhibitors | Yes | 36 | 30 | 0.559 |
| | No | 19 | 11 | |
| Carbapenems | Yes | 12 | 10 | 0.959 |
| | No | 43 | 31 | |
| Quinolones | Yes | 6 | 2 | 0.461 |
| | No | 49 | 39 | |
| Gram-positive antibiotics | Yes | 8 | 12 | 0.075 |
| | No | 47 | 29 | |
| Hypertension | Yes | 28 | 26 | 0.311 |
| | No | 27 | 15 | |
| Diabetes | Yes | 11 | 16 | 0.069 |
| | No | 44 | 25 | |
| Central venous catheter | Yes | 46 | 37 | 0.526 |
| | No | 9 | 4 | |
| Enteral nutrition | Yes | 29 | 30 | 0.068 |
| | No | 26 | 11 | |
| Parenteral nutrition | Yes | 15 | 15 | 0.453 |
| | No | 40 | 26 | |
| Vasoactive drugs | Yes | 17 | 9 | 0.456 |
| | No | 38 | 32 | |

[a]Indicates $P < 0.05$.

## Potential transmission of ESBL-EC

To study the spread of ESBL-EC in the ICU, we analyzed the SNP differences among 82 strains of ESBL-EC and defined SNP ≤22 as a homologous strain. There were minimal differences in SNPs among 24 strains of ESBL-EC in the same ST type, suggesting an existing potential transmission relationship (Fig. 5).

Based on the results of the SNP analysis (Fig. 6), we considered that the ESBL-EC obtained from the isolation of DY77 (P1), DY116 (P2), and DY313 (P3) of ST10 were the homologous strain group. Furthermore, DY156 (P4), DY172 (P4), DY177 (P5), DY178 (P6), DY188 (P6), DY325 (P7), and DY326 (P4) of ST10 were considered to be another homologous strain group. Considering the difference in sample acquisition time and position, it is suggested that the spread and distribution of ESBL-EC in the ICU were caused by the movement of medical personnel. For example, DY77 detected from the

**TABLE 2** Multivariate analysis of high-risk factors of ESBL-EC colonization in ICU patients

|  | Odds ratio (OR) | 95.0% CI Exp(B) | P |
| --- | --- | --- | --- |
| Enteral nutrition | 4.358 | 1.193–18.799 | 0.034[a] |
| History of third-generation cephalosporin exposure | 25.054 | 2.206698.842 | 0.020[a] |
| History of special class of anti-positive bacterial drugs exposure | 6.717 | 1.30942.892 | 0.030[a] |
| Albumin | 0.847 | 0.7170.983 | 0.036[a] |

[a]Indicates $P < 0.05$.

anal swab of patient 1 (P1) in the first bed during the eighth week of the study was the same as DY116 detected from the anal swab of P2 in the 12th bed during the 11th week. Additionally, both were homologous to the DY313 detected in the stethoscope in the surrounding environment of P3 in bed 19th during the ninth week.

Furthermore, DY135 (P8) and DY321 (P9) of ST38 from bed 20 and 25, respectively, were considered to be the homologous strains. Both were relatively far apart, and DY135 was an anal swab from the patient, whereas DY321 was from an overflow mouth wall, suggesting that their transmission might be due to mobile hand washing by healthcare workers.

Notably, DY125 (P10), DY138 (P10), DY168 (P10), DY169 (P10), DY189 (P6), and DY324 (P10) of ST95 were identified as homologous strains. DY125, DY138, DY168, DY169, and DY324 were isolated from the same patient (P10) in bed 27 during the 12th and 13th week and from a patient in bed 4 during the 16th week. Meanwhile, DY189 was isolated from another patient (P6) in bed 24 during the 17th week. As shown in the green icon of Fig. 5, their transmission was similarly considered to be caused by healthcare worker turnover.

As shown in the pink icon of Fig. 5, DY84 (P11), DY100 (P12), DY101 (P12), and DY102 (P12) from ST224 were identified as homologous strains, and their transmission was also considered to be caused by healthcare workers' mobility. Furthermore, as shown in the sky-blue icon, DY20 (P13) and DY158 (P14) of ST1193 were patient samples from beds 16 and 15, respectively, and all of the samples were from patients in adjacent beds in the same room, suggesting that transmission was possibly caused by the shared

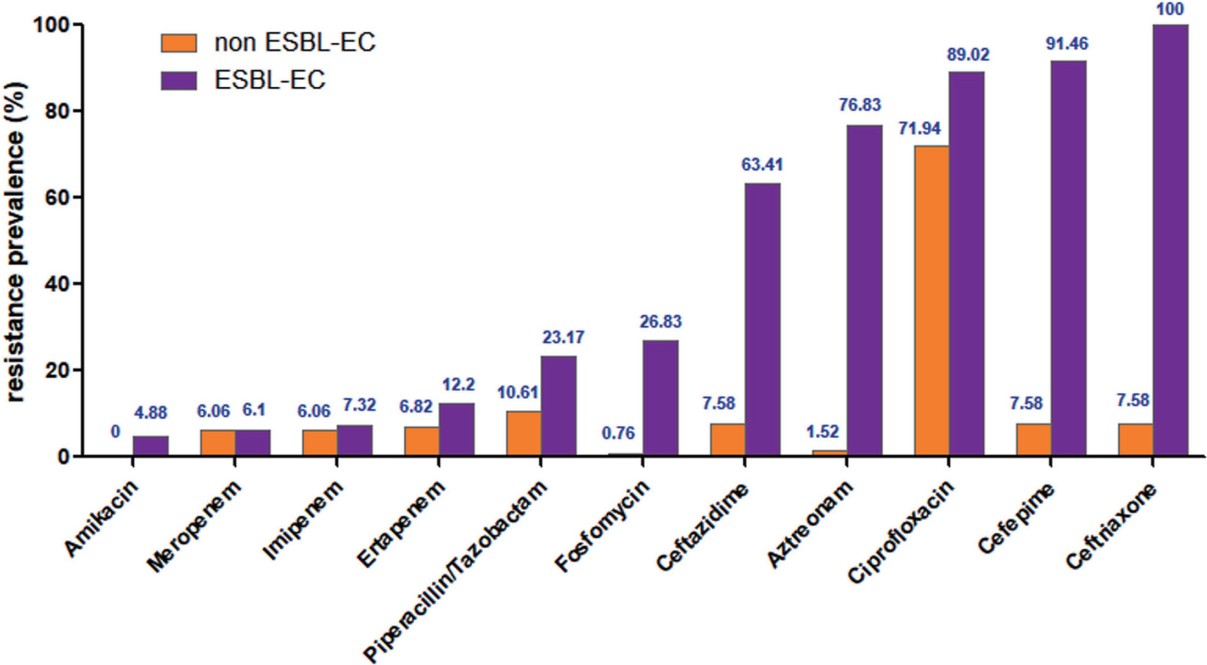

**FIG 2** Antimicrobial resistance rates of 214 *E. coli* strains.

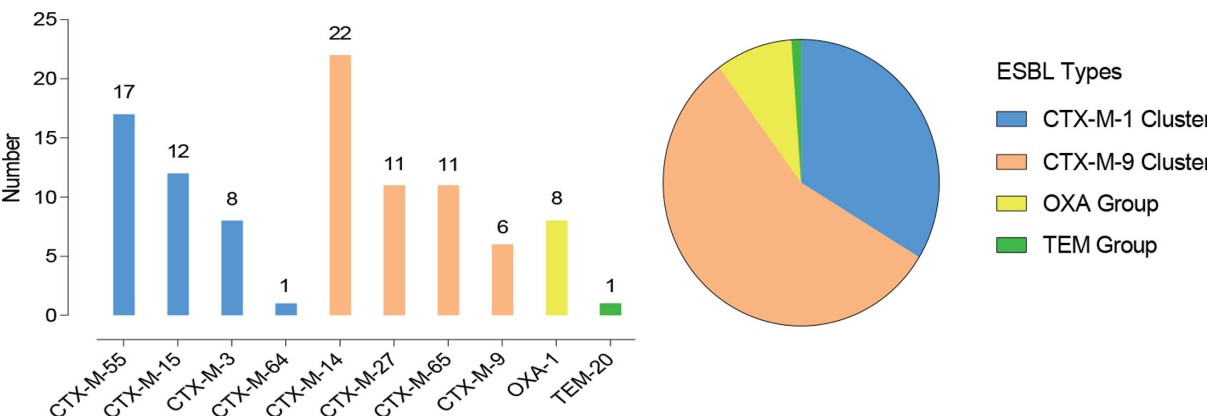

**FIG 3** Distribution of ESBL genotypes in 82 ESBL-EC strains.

environment in large rooms. In summary, personnel were involved in the transfer of the isolates.

## DISCUSSION

The gut microbiota contributes significantly to endogenous infections among patients in the ICU (24–26). Owing to the rising incidence of infections caused by ESBL-producing gram-negative bacteria in clinical settings (27, 28), the risk factors associated with patients' intestinal colonization need to be urgently identified. The findings from this study have revealed that exposures to specific classes of drugs against gram-positive bacteria and third-generation cephalosporins are associated with the risk factors for ESBL-EC colonization. The normal gut microbiota is a barrier against the overgrowth of existing opportunistic pathogens and colonization by potential pathogens. However, the use of antibacterial drugs, particularly in patients with severe disease conditions or a compromised immune system, can disturb the ecological balance between the host and the normal microbiota, thereby disrupting their control over the growth of opportunistic microorganisms and the host's resistance to their colonization (29). Moreover, the use of antibiotics creates a selection pressure that prioritizes the bacterial strains with resistant genes, allowing multidrug-resistant (MDR) pathogens to accumulate in the gut (30). A study involving patients in the ICU showed that advanced age and exposure to piperacillin/tazobactam and vancomycin were the risk factors for intestinal colonization by ESBL-producing gram-negative bacteria (31). Another study involving neonates in the ICU revealed that a lower birth weight, a younger gestational age, and exposure to vancomycin were the risk factors for intestinal colonization by this bacteria (32). Reportedly, exposure to third-generation cephalosporins is an important risk factor for ESBL-EC infections (33, 34). Additionally, findings from a prospective study of community-acquired urinary tract infections in China revealed that the overall positive infectious rate of ESBL-producing gram-negative bacteria among the *Enterobacteriaceae* was 37.2% (562/1512), and the use of cephalosporins within 3 months was an independent risk factor for colonization by ESBL-producing bacteria that trigger such infections (35). These previous studies support the conclusion of our study that specific classes of drugs against gram-positive bacteria and exposure to third-generation cephalosporins correlate with intestinal colonization by ESBL-EC. Therefore, patients who are exposed to these drugs should be carefully monitored in clinical settings, and relevant and effective measures should be promptly taken when appropriate.

Furthermore, we found that the use of nasogastric tubes and reduced albumin levels were also risk factors for ESBL-EC colonization. Enteral feeding provides a site for bacterial colonization and appropriate growth conditions for ESBL-EC development, thereby promoting intestinal colonization of the pathogen (36). Reportedly, nasogastric tubes used for enteral feeding can provide sites for colonization by biofilm-producing

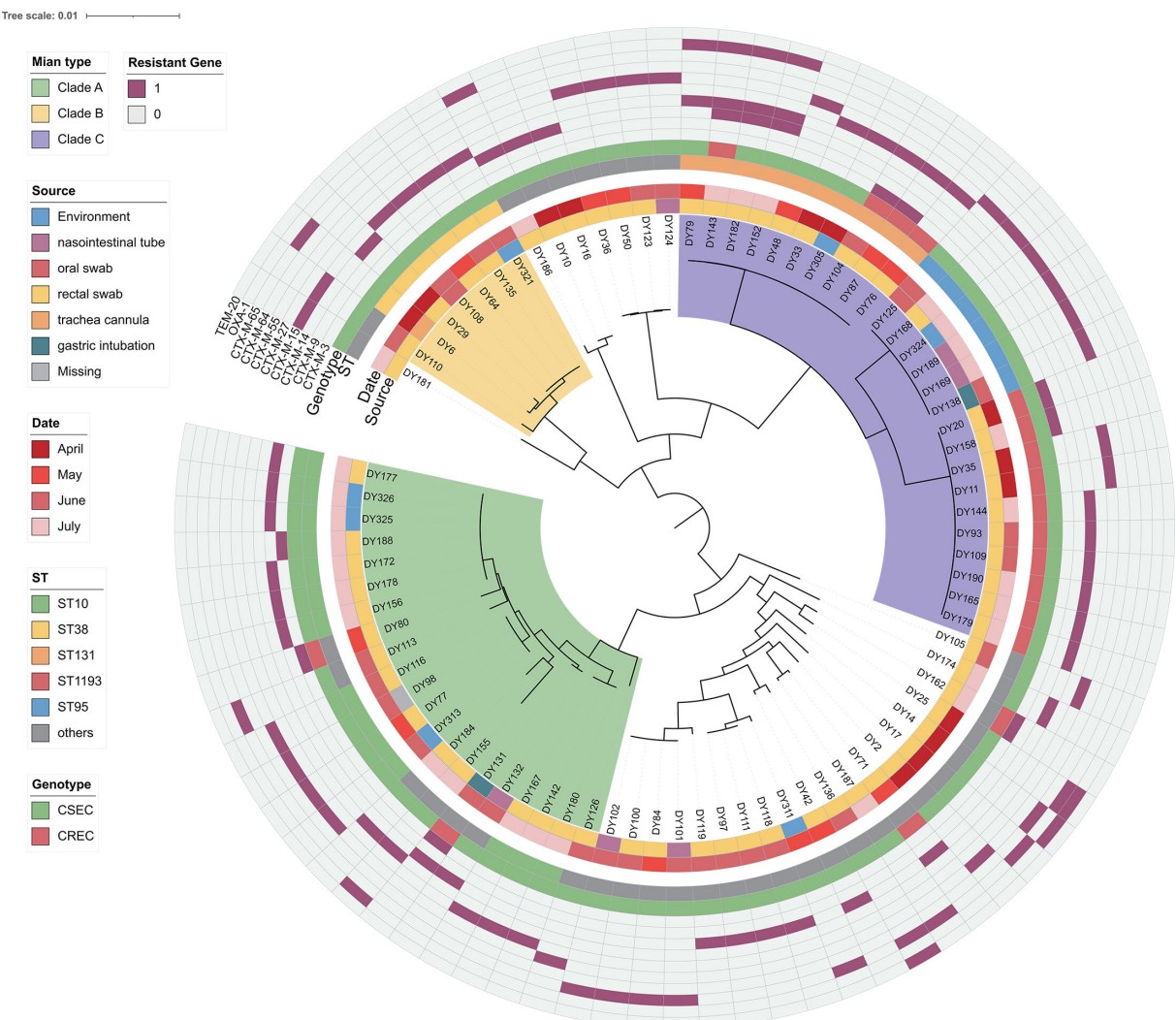

**FIG 4** Phylogenetic tree of 82 ESBL-EC strains. The phylogenetic tree constructed by the maximum likelihood method reflects the genetic relationships among all 82 ESBL-EC strains. From inside to outside, the main branches (divided into three branches), collection time, sample source, sequence type (ST), genotype, and ESBL enzyme type for all ESBL-EC strains are successively listed in each layer.

bacteria, such as *Enterobacteriaceae*, including ESBL-EC (37). Bacteria can enter the gastrointestinal tract in bacterial biofilm clumps during each tube-feeding event. Hence, this feeding tube is a risk factor for infections and a source of pathogen colonization (38, 39). Moussa et al. (40) identified enteral feeding tubes as one of the independent risk factors for colonization by ESBL-producing *Enterobacteriaceae*. Kajihara et al. (41), in their cohort study, also indicated that ESBL-EC infection was associated with enteral feeding. Furthermore, bacterial colonization in feeding tubes can influence intestinal bacterial colonization. Ogrodzki et al. (42) discovered significant persistent colonization by the same bacterial strain in both the feeding tube and the intestines of patients. Therefore, the use of feeding tubes may contribute to patients being persistent hosts for the continuous transmission of ESBL-EC, exposing the individuals to the risks of prolonged colonization by bacterial strains and recurrent infections (43). At present, only a few studies exist on the correlation between reduced albumin levels and bacterial colonization. Ikeda et al. (44) found that serum albumin levels and lymphocyte counts were significantly lower in patients with ESBL-EC-induced invasive infections than in those without. Therefore, low serum albumin may be a risk factor for ESBL-EC infections.

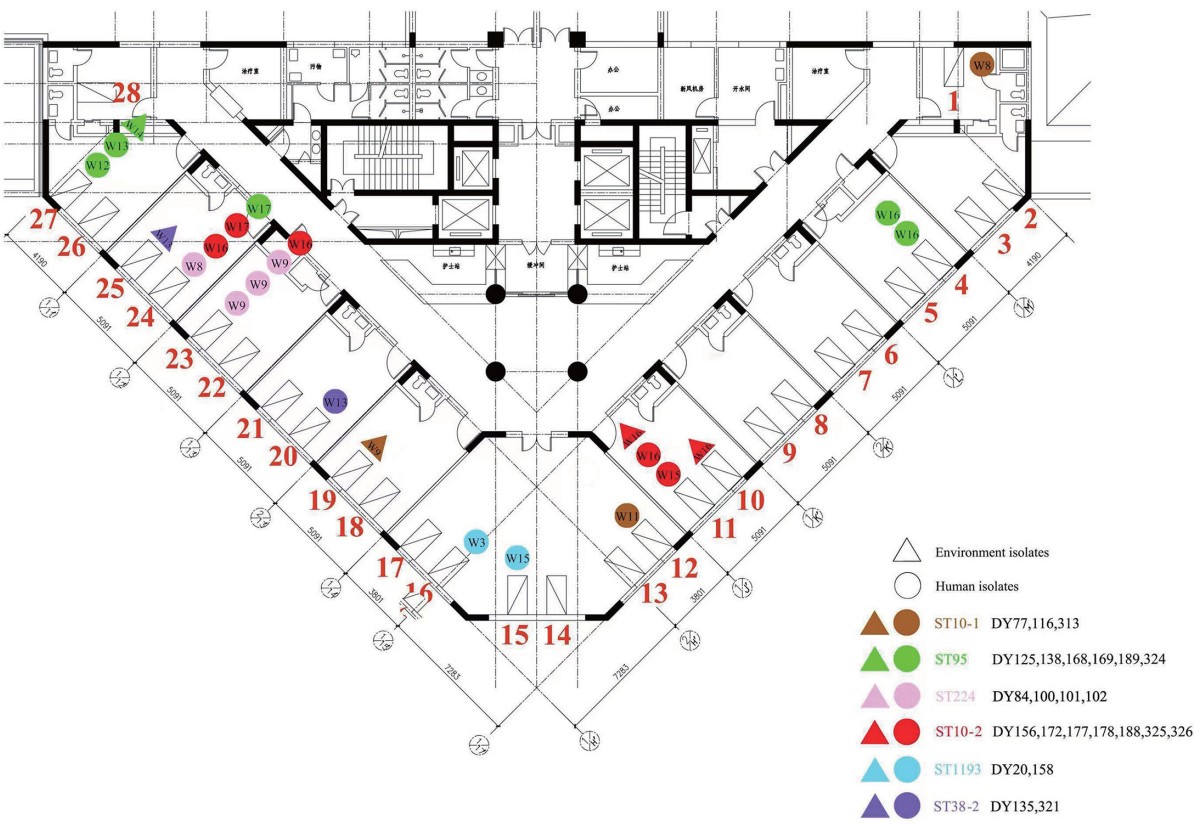

**FIG 5** ICU floor sketch map and overall distribution of ESBL-EC strains. The ICU on the fourth floor of the hospital has a total of 28 beds, including 2 single rooms (Nos. 1 and 28), 1 six-bed room (Nos. 12, 13, 14, 15, 16, and 17), and 10 two-bed rooms (remaining rooms). Strains from the same bed can be a superposition of ESBL-EC positive results collected at different times. Clinical samples are represented by circles and environmental samples are represented by triangles. Different colors represent ESBL-EC strains with different SNP differences (>22). The numbers in the triangle or circle represent the weeks of strain screening. ST10-1 and ST10-2 were both ST10-type ESBL-EC. ST38-1 and ST38-2 were also ST10-type ESBL-EC, which can be distinguished by the SNP difference (>22).

In our study, the colonization rate of acquired ESBL-EC in the ICU was 41.46% (17/41), which was lower than that in a previous study that showed that the acquisition rate of ESBL-EC in the ICU was 63.8% (236/370) and the median time for patients entering the ICU to being infected with ESBL-EC was 10 days (IQR = 6–16) (45). The median time in our study was 13 days (IQR = 8-41), which was similar to the previous study. The CTX-M types of ESBL-EC in this study were dominated by the $bla_{CTX-M-1}$ and $bla_{CTX-M-9}$ clusters, along with other types, such as the OXA and TEM clusters. Shi et al. (46) found that the $bla_{CTX-M-1}$ and $bla_{CTX-M-9}$ groups of the Chinese ESBL-positive *E. coli* isolates were the major CTX-M gene types. CTX-M-14 was the most abundant ESBL detected in this study, which was consistent with the findings from Zhong et al. (47) in Hunan, China. The second most frequent type found in this study was CTX-M-55. In addition, Bevan et al. (27) showed that the isolation rate of *Enterobacteriaceae*-producing bacteria with CTX-M-55 has increased substantially in recent years in China. Similarly, Zeng et al. (48) in Guangzhou, China, also concluded that the percentage of $bla_{CTX-M-55}$ positive *E. coli* significantly exceeded that of $bla_{CTX-M-15}$. Moreover, no known ESBL-encoding genes were identified in the two ESBL-positive *E. coli* strains, suggesting the possible existence of novel ESBL genes. Further studies are needed to investigate these mechanisms. The results of the MLST analysis showed that the major ST type of *E. coli* in this study was ST10, followed by ST131 and ST1193. ST10 is a base sequence containing several resistance genes, such as $bla_{CTX-M-15}$ (49). ST131, the second most abundant ST type identified in this study, is a recently emerged and globally disseminated clonal strain. It also contributes mainly to hospital and community-acquired urinary tract infections as

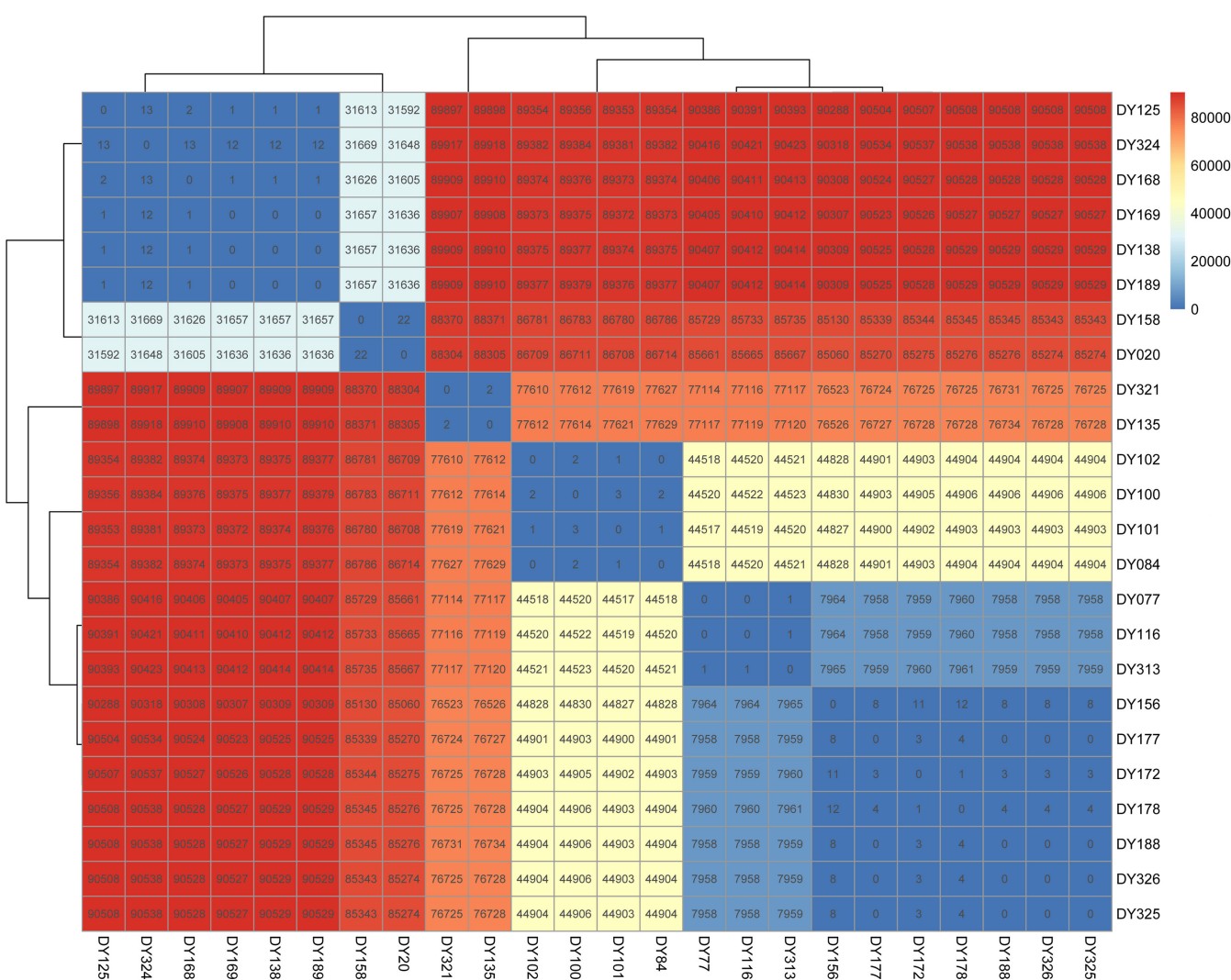

**FIG 6** Heat map of SNP differences between 24 ESBL-EC strains with transmission relationships.

well as bloodstream, pets, and poultry infections (50). ST1193 was also detected in the study as the second most frequent ST type, an emerging global MDR high-risk clone, and an important cause of community-onset urinary and bloodstream infections (51).

We found that ESBL-EC strains can be transmitted from patient to environment, with at least eight transmission events observed, of which six may have been caused by healthcare workers' mobility and two by the spread of the infection in the environment. These findings reveal that rather than investing in additional single-room units, better training and management of healthcare standard precautions are more important. These precautionary measures include the use of gloves, any other necessary barriers prior to contact with body fluids, wounds, and mucous membranes, and hand hygiene measures in accordance with the WHO five-step hand hygiene method (52). Contact isolation is defined as isolating oneself when caring for patients infected with anti-microbial-resistant bacteria and implementing precautions, such as wearing gloves and isolation gowns during all interactions with patients and their immediate environment, staying in single-bed rooms, or grouping in multi-bed rooms (53, 54). Previous reports show that in single-bed rooms with a relatively low prevalence of ESBL-EC and high compliance with standard precautions, the incidence of ICU-acquired ESBL-EC colonization and infection did not significantly differ after complete discontinuation of contact isolation measures, suggesting that single-bed room isolation measures are not superior to those

of standard precautions (16). Another study showed that a low cross-transmission rate of ESBL-EC could be achieved while ensuring hand hygiene compliance among healthcare workers (2). Tschudin-Sutter et al. (55) found that ESBL-EC may not survive as long as other gram-negative bacteria on environmental surfaces, possibly explaining why contact prophylaxis did not significantly reduce the ESBL-EC transmission rate. Kluyt-mans-van den Bergh et al. (56) showed that multi-bed wards with contact isolation measures were not inferior to single-room wards with the same measures in preventing transmission of ESBL *Enterobacteriaceae*. Furthermore, a cost-effectiveness analysis showed that improving hand hygiene compliance was the most cost-effective strategy to prevent ESBL-EC transmission, while screening and exposure prophylaxis was the least effective strategy (57). This suggests that more attention should be given to the training of healthcare workers in hand hygiene and other related measures to reduce ESBL-EC colonization and transmission.

## Conclusion

In conclusion, our findings revealed that a high percentage of colonized ESBL-EC was found in the ICU, most of which were acquired from the hospital. CTX-M-14, CTX-M-55, and CTX-M-15 were the most common genotypes of ESBL-EC, and ST10, ST131, and ST1193 were the most predominant ST clones. Enteral nutrition, history of third-generation cephalosporin exposure, a special class of anti-positive bacteria drug exposure, and albumin are high-risk factors for ESBL-EC colonization. Furthermore, ESBL-EC can be clonally transmitted from patient to environment, with the movement of healthcare workers being the most common transmission. More effective interventions and active screening are needed to prevent and control ESBL-EC colonization in clinical settings.

## ACKNOWLEDGMENTS

This study was supported by the Key Research and Development Program from the Ministry of Science and Technology of China (Grant no. 2023YFC2307100).

Y.D., Y.J., and Y.S.Y. conceived and drafted the manuscript. M.J.S and Q.X.P. collected and identified the strain. Y.Y., L.J.X., and J.T.H. performed antimicrobial susceptibility testing. J.X.Z., H.Y.L., and Y.J.L. performed the ESBL phenotypic confirmation experiments. M.L. and X.L.G. collected Clinicalclinical data. H.M.Z. performed the whole-genome sequencing and bioinformatic analysis. All authors reviewed the manuscript.

## AUTHOR AFFILIATIONS

[1]Department of Intensive Care Unit, Sir Run Run Shaw Hospital Qiantang Campus, Zhejiang University School of Medicine, Hangzhou, Zhejiang, China
[2]Key Laboratory of Microbial Technology and Bioinformatics of Zhejiang Province, Hangzhou, Zhejiang, China
[3]Regional Medical Center for National Institute of Respiratory Diseases, Sir Run Run Shaw Hospital, Zhejiang University School of Medicine, Hangzhou, Zhejiang, China
[4]Department of Infectious Diseases, Sir Run Run Shaw Hospital, Zhejiang University School of Medicine, Hangzhou, Zhejiang, China
[5]Department of Pharmacy, Sir Run Run Shaw Hospital Qiantang Campus, Zhejiang University School of Medicine, Hangzhou, Zhejiang, China
[6]Zhejiang University School of Medicine, Hangzhou, Zhejiang, China

## AUTHOR ORCIDs

Ying Ding  http://orcid.org/0000-0002-5349-2163
Yunsong Yu  http://orcid.org/0000-0003-2903-918X
Yan Jiang  http://orcid.org/0000-0002-5877-9286

## FUNDING

| Funder | Grant(s) | Author(s) |
|---|---|---|
| Key Technologies Research and Development Program | 2023YFC2307100 | Yunsong Yu |

## DATA AVAILABILITY

All genome data have been uploaded to NCBI under BioProject accession number PRJNA911045.

## ETHICS APPROVAL

This study was reviewed and approved by the Ethics Committee of Sir Run Run Shaw Hospital, Zhejiang University School of Medicine (No. 20201217-33).

## ADDITIONAL FILES

The following material is available online.

### Open Peer Review

**PEER REVIEW HISTORY (review-history.pdf).** An accounting of the reviewer comments and feedback.

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
