## [Reviewer comments · Microbiology Spectrum]

Microbiology Spectrum

Genetic Characteristics and Risk Factors of Extended-spectrum β -lactamase-producing *Escherichia coli* in Chinese Intensive Care Unit: a Prospective Molecular Epidemiology Study

Ying Ding, Meijun Song, Yi Yang, Hemu Zhuang, Qiuxiang Pan, Lijie Xu, Jintao He, Junxin Zhou, Haiyang Liu, Yongjun Lin, Min Liang, Xiuli Guo, Yunsong Yu, and Yan Jiang

Corresponding Author(s): Yan Jiang, Zhejiang University School of Medicine Sir Run Run Shaw Hospital

Review Timeline:

Submission Date:	November 24, 2024
Editorial Decision:	February 20, 2025
Revision Received:	May 14, 2025
Accepted:	June 13, 2025

Editor: Rosemary She

Reviewer(s): Disclosure of reviewer identity is with reference to reviewer comments included in decision letter(s). The following individuals involved in review of your submission have agreed to reveal their identity: Kai Zhou (Reviewer #1)

Transaction Report:

DOI: <https://doi.org/10.1128/spectrum.02895-24>

Re: Spectrum02895-24 (**Genetic Characteristics and Risk Factors of Extended-spectrum β -lactamase-producing *Escherichia coli* in Chinese Intensive Care Unit: a Prospective Molecular Epidemiology Study**)

Dear Mr. Yan Jiang:

Thank you for the privilege of reviewing your work. Below you will find my comments, instructions from the Spectrum editorial office, and the reviewer comments.

Revision Guidelines

Sincerely,
Rosemary She
Editor
Microbiology Spectrum

Reviewer #1 (Comments for the Author):

The manuscript submitted by Ding et al. conducts a 4-month prospective observational study in 2021 aimed at analyzing the epidemiological and genetic characteristics of ESBLs-EC in ICU patients and the hospital environment. In this study, the authors analyzed the major ESBLs types and predominant STs of 82 ESBLs-EC strains isolated from 214 *E. coli* strains. Additionally, multivariate logistic regression was used to analyze high-risk factors for ESBLs-EC colonization. These findings are helpful for guiding clinicians in the rational selection of antibiotics and for preventing and controlling ESBLs-EC through effective

interventions and proactive screening. This work is interesting and relevant, providing appropriate evidence and explanations. However, to further improve the manuscript, the following suggestions should be considered:

1. The authors should clearly specify the sources of the 214 E. coli strains (both sensitive and resistant) to show how many strains were isolated from patients and environmental sources.
2. In Figure 3, the color of the data for May and June is too subtle. I suggest to change the colors for better distinction.
3. Line 152: "MLST was determined by the PubMLST dataset." More methodological details should be included. Please specify the MLST scheme used, i.e. the Achtman scheme or the Pasteur scheme?
4. Please provide the reference for the clade assignment (clades A, B, C, D) of E. coli.
5. Since this is a prospective observational study, it must state that whether the study has been approved by the local ethics committee or not.
6. The method used for patient screening is unclear. The inclusion criteria and screening frequency should be described. It is currently unclear whether all patients were screened upon admission (within 48 hours, see Line 187). How did the authors define the exact method for patients hospitalized from day 2 to day 7?
7. It is unclear how the authors distinguished between clades and ST types.
8. All genome sequences should be deposited, and the accession numbers should be included in the main text.

Reviewer #2 (Comments for the Author):

Spectrum02895-24: Genetic Characteristics and Risk Factors of Extended-spectrum β -lactamase-producing *Escherichia coli* in Chinese Intensive Care Unit: a Prospective Molecular Epidemiology Study

This study is an observational study linking ESBL-producing E.coli isolated from patients, with the type of ESBL and sequence type of the E. coli with the site of collection and the location of the patient in the hospital. The idea is interesting, but the authors did not go far enough to make the paper that interesting. Although they tried to link the location of the collection site (mostly anal) with the rooms of the hospital with the environmental source but only one isolate was linked to an environmental source, a stethoscope.

As written the manuscript is not very interesting. It would be a much better paper if environmental links (not just the rooms of the patient) could be definitively identified. Some of the data, such as risk factors have already been noted by other publications so there are no new information there.

Figure 5 was interesting, but it just doesn't go far enough in the ability to link the environmental sources of the isolates found in the patients. Although it suggests that personnel had been involved in transfer of isolates it was not definitively shown.

Two major problems of this manuscript are:

1. There are no references cited within the paper. References are given but no citations are found in the text.
2. Grammar and writing style need to be edited by a person fluent in English.

Response Letter

Dear editor,

I wish to re-submit the manuscript titled “**Genetic Characteristics and Risk Factors of Extended-spectrum β -lactamase-producing *Escherichia coli* in Chinese Intensive Care Unit: A Prospective Molecular Epidemiology Study**”. The manuscript ID is Spectrum02895-24R1.

We thank you and the reviewers for your thoughtful suggestions and insights. The manuscript has benefited from these insightful suggestions. I look forward to working with you and the reviewers to move this manuscript closer to publication in *Microbiology Spectrum*.

The manuscript has been rechecked and the necessary changes have been made in accordance with the reviewers’ suggestions. Our detailed responses to the reviewers’ comments can be found as follows. The comments from the reviewers or editor are in black, and responses from the authors are in blue. Revisions to the manuscript text are indicated in yellow.

Thank you for your consideration. I look forward to hearing from you.

Sincerely,

Yunsong Yu

Tel: +86-571-8600-6142;

Fax: +86-571-8600-6142

Email: yvys119@zju.edu.cn,

Yan Jiang

Tel: +86-571-8600-6142

Fax: +86-571-8600-6142

Email: jiangy@zju.edu.cn

Response to Reviewer #1 comments

The manuscript submitted by Ding et al. conducts a 4-month prospective observational study in 2021 aimed at analyzing the epidemiological and genetic characteristics of ESBLs-EC in ICU patients and the hospital environment. In this study, the authors analyzed the major ESBLs types and predominant STs of 82 ESBLs-EC strains isolated from 214 *E. coli* strains. Additionally, multivariate logistic regression was used to analyze high-risk factors for ESBLs-EC colonization. These findings are helpful for guiding clinicians in the rational selection of antibiotics and for preventing and controlling ESBLs-EC through effective interventions and proactive screening. This work is interesting and relevant, providing appropriate evidence and explanations. However, to further improve the manuscript, the following suggestions should be considered:

Comment 1

1. The authors should clearly specify the sources of the 214 *E. coli* strains (both sensitive and resistant) to show how many strains were isolated from patients and environmental sources.

Response: Thanks for your helpful advice. We have provided the information in the Results part (Line 177-179). Line 177-179: “A total of 214 *E. coli* strains were isolated, of which 191 were isolated from 99 different patients and 23 were isolated from the surrounding environment of patients in the ICU ward.”

Comment 2

2. In Figure 4, the color of the data for May and June is too subtle. I suggest to change the colors for better distinction.

Response: Thank you for your careful review and useful suggestion. As suggested, we have changed the color of the data for May and June in Figure 4.

Figure 4 Distribution of ESBL genotypes in the 82 ESBL-EC strains.

Comment 3

3. Line 152: "MLST was determined by the PubMLST dataset." More methodological details should be included. Please specify the MLST scheme used, i.e. the Achtman scheme or the Pasteur scheme?

Response: Thank you for your suggestion. The MLST uses the Achtman scheme. We have provided the information in the Materials and Methods part (Line 154-156). Line 154-156: "The MLST used in the survey was the Achtman scheme, which was performed using mlst (<https://github.com/tseemann/mlst>) with the PubMLST dataset."

Comment 4

4. Please provide the reference for the clade assignment (clades A, B, C, D) of *E. coli*.

Response: Thank you for your suggestion. In our study, we determined the *E. coli* clade based on the topology structure in the phylogenetic tree, which is represented

the relative distant relationship of the strains. There is no unified clustering method for reference.

Comment 5

5. Since this is a prospective observational study, it must state that whether the study has been approved by the local ethics committee or not.

Response: Thank you for your suggestion. We have an ethics approval and it can be found in the revised manuscript on Line 147-149. Line 147-149: “This study was reviewed and approved by the Ethics Committee of Sir Run Run Shaw Hospital, Zhejiang University School of Medicine (No. 20201217-33).”

Comment 6

6. The method used for patient screening is unclear. The inclusion criteria and screening frequency should be described. It is currently unclear whether all patients were screened upon admission (within 48 hours, see Line 187). How did the authors define the exact method for patients hospitalized from day 2 to day 7?

Response: Thanks for your comment. The following were the inclusion criteria: all of the patients admitted to the ICU during the study period, sampled collected including oropharyngeal, rectal, tracheal intubation, gastric intubation, tracheotomy tube, and nasointestinal tube swabs, which can be found at Line 137-140. The method of screening patients is within 4 months (2021.4.1-2021.7.31), screening every Tuesday morning, with the inclusion criteria including 6 sites for all patients, and 19 environmental points around the patient. We did not screen the patients within 48 hours of admission, we did the statistics analysis based on the medical records. Admission to ICU 24-48 hours counts as the second day, 48-72 hours counts as the third day and so on. The seventh day is 144-168 hours of hospitalization. The hospitalization time could be collected in the medical record system and calculated accurately.

Comment 7

7. It is unclear how the authors distinguished between clades and ST types.

Response: We constructed the phylogenetic tree using core genes of the strains. The clade classification was determined based on the structure of this phylogenetic tree, meaning that the clade delineation was fundamentally rooted in the core genome of the strains. For ST (Sequence Type) classification, we followed the standard *E. coli* typing scheme based on seven housekeeping genes according to the PubMLST dataset. Generally, strains sharing the same ST are likely to belong to the same clade.

Comment 8

8. All genome sequences should be deposited, and the accession numbers should be included in the main text.

Response: Thank you for your suggestion. We have added the accession number in Line 402-403. Line 402-403: "All genome data have been uploaded to NCBI under BioProject accession number PRJNA911045."

Response to Reviewer #2 comments

This study is an observational study linking ESBL-producing *E. coli* isolated from patients, with the type of ESBL and sequence type of the

with the site of collection and the location of the patient in the hospital. The idea is interesting, but the authors did not go far enough to make the paper that interesting. Although they tried to link the location of the collection site (mostly anal) with the rooms of the hospital with the environmental source but only one isolate was linked to an environmental source, a stethoscope.

As written the manuscript is not very interesting. It would be a much better paper if environmental links (not just the rooms of the patient) could be definitively identified. Some of the data, such as risk factors have already been noted by other publications so there are no new information there.

Figure 5 was interesting, but it just doesn't go far enough in the ability to link the environmental sources of the isolates found in the patients. Although it suggests that

personnel had been involved in transfer of isolates it was not definitively shown.

Response: Thank you for your careful review and useful suggestion. The purpose of our project is to screen for ESBL-producing *E. coli* from patient samples and the environment of the rooms where patients are located, due to the strong evidence suggesting that the ward environment plays a crucial role as a key medium for the rapid transmission of multidrug resistant bacteria between patients. As shown in the figure below, the patient samples we screen come from six anatomical sites, including anal swabs, throat swabs, nasointestinal tubes, nasogastric tubes, tracheal intubation tubes, and tracheostomy tubes. The environmental samples from wards are collected from 19 sites, including bed rails, ventilator button panels, ECG monitors, infusion pumps, light switch buttons, nebulizers, stethoscopes, ceiling booms, bed regulators, bedside cabinets, computer mice and keyboards, IV stand hooks, treatment carts, storage cabinets, internal sink drain pipes, faucet surfaces, sink countertops, and internal overflow holes and water in the drainage channel.

As illustrated in Figure 5, the Δ triangle symbols represent the samples from environment, while the \circ circular symbols represent samples from patient. As a key finding of our study, we identified multiple instances of ESBL-EC transmission between patient and environmental samples, not just in the case of the stethoscope-associated strain. For instance, anal swab samples DY156 and DY172 from bed 11 were transmitted to bed rail sample DY326 (red mark in Figure 5). Pharyngeal swab sample DY125 and nasogastric tube sample DY138 from bed 27 were transmitted to bedside cabinet sample DY324 (green mark in Figure 5), then to

nasointestinal tube sample DY189 of another patient in bed 24 three weeks later. Anal swab sample DY135 from bed 20 was transmitted to the overflow hole wall sample DY321 of bed 25 (purple mark in Figure 5). These results showed that ESBL-EC transmission was mainly driven by personnel movement. As suggested, we have explicitly stated that personnel were involved in the transfer of the isolates in Line 275.

Figure 5 ICU floor sketch map and overall distribution of ESBL-EC strains. The ICU on the fourth floor of the hospital has a total of 28 beds, including two single rooms (Nos. 1 and 28), 1 six-bed room (Nos. 12, 13, 14, 15, 16, and 17) and 10 two-bed rooms (remaining rooms). Strains from the same bed can be a superposition of ESBL-EC positive results collected at different times. Clinical samples are represented by circles and environmental samples are represented by triangles. Different colors represent ESBL-EC strains with different SNP differences (> 22). The numbers in the triangle or circle represent the weeks of strain screening. ST10-1 and ST10-2 were both ST10-type ESBL-EC. ST38-1 and ST38-2 were also ST10-type ESBL-EC, which can be distinguished by the SNP difference (> 22).

In addition, our prospective observational study specifically investigated the risk factors for ESBL-EC colonization, in contrast to previously published retrospective

studies focusing on ESBL-EC infection risks. Therefore, our findings provided novel evidence and perspectives for the development of epidemiological prevention and control strategies targeting ESBL-EC.

Comment 1

1. There are no references cited within the paper. References are given but no citations are found in the text.

Response: Thank you for your careful review. We have added the citations in the revised manuscript.

Comment 2

2. Grammar and writing style need to be edited by a person fluent in English.

Response: Thank you for your suggestion. As suggested, we have engaged fluent English speakers to refine the manuscript's grammar and writing style. The modified parts in the revised manuscript are highlighted in yellow. Additionally, a certificate from the professional editing company is provided below for verification purposes.

Editing Certificate

This document certifies that the manuscript listed below has been edited to ensure language and grammar accuracy and is error free in these aspects. The logical presentation of ideas and the structure of the paper were also checked during the editing process. The edit was performed by professional editors at Editage, a brand of Cactus Communications. The author's core research ideas were not altered in any way during the editing process. The quality of the edit has been guaranteed, with the assumption that our suggested changes have been accepted and the text has not been further altered without the knowledge of our editors.

MANUSCRIPT TITLE

Genetic Characteristics and Risk Factors of Extended-spectrum β -lactamase-producing Escherichia coli in Chinese Intensive Care Unit: A Prospective Molecular Epidemiology Study

AUTHORS

YING DING

ISSUED ON

March 17, 2025

JOB CODE

HHTWY_5_3

Prabh Grewal
Senior Vice President - Editage

editage | helping you
get published

Since 2002, Editage has helped over 430,000 authors publish around 1.2 million research papers in scholarly journals across over 1000 disciplines through editorial, translation, transcription, and publication support services. Editage is a brand of Cactus Communications (cactusglobal.com), a science communication and technology company.

GLOBAL :
+1(669) 272-1214 | request@editage.com

CHINA :
400-001-8237; 021-60209400 |
fabiao@editage.cn

CACTUS

Re: Spectrum02895-24R1 (**Genetic Characteristics and Risk Factors of Extended-spectrum β -lactamase-producing *Escherichia coli* in Chinese Intensive Care Unit: a Prospective Molecular Epidemiology Study**)

Dear Mr. Yan Jiang:

Your manuscript has been accepted, and I am forwarding it to the ASM production staff for publication. Your paper will first be checked to make sure all elements meet the technical requirements. ASM staff will contact you if anything needs to be revised before copyediting and production can begin. Otherwise, you will be notified when your proofs are ready to be viewed.

Sincerely,
Rosemary She
Editor
Microbiology Spectrum